# Evaluation of Antidiabetic Effect of Luteolin in STZ Induced Diabetic Rats: Molecular Docking, Molecular Dynamics, In Vitro and In Vivo Studies

**DOI:** 10.3390/jfb14030126

**Published:** 2023-02-25

**Authors:** Ozair Alam, Lamya Ahmed Al-Keridis, Jalaluddin Khan, Sameena Naaz, Afshar Alam, Syed Amir Ashraf, Nawaf Alshammari, Mohd Adnan, Md Amjad Beg

**Affiliations:** 1Department of Computer Science & Engineering, School of Engineering Sciences and Technology (SEST), Jamia Hamdard University, New Delhi 110062, India; 2Medicinal Chemistry & Molecular Modelling Lab, Department of Pharmaceutical Chemistry, School of Pharmaceutical Education and Research (SPER), Jamia Hamdard, New Delhi 110062, India; 3Department of Biology, College of Science, Princess Nourah bint Abdulrahman University, P.O. Box 84428, Riyadh 11671, Saudi Arabia; 4Microbial & Pharmaceutical Biotechnology Laboratory (MPBL), Department of Pharmacognosy & Phytochemistry, School of Pharmaceutical Education and Research (SPER), Jamia Hamdard, New Delhi 110062, India; 5Department of Clinical Nutrition, College of Applied Medical Science, University of Ha’il, Ha’il P.O. Box 2440, Saudi Arabia; 6Department of Biology, College of Science, University of Ha’il, Ha’il P.O. Box 2440, Saudi Arabia; 7Centre for Interdisciplinary Research in Basic Science, Jamia Millia Islamia, Jamia Nagar, New Delhi 110025, India

**Keywords:** luteolin, diabetes mellitus, oxidative stress, inflammation, molecular docking, molecular dynamic simulation

## Abstract

Despite the existence of modern antidiabetic medications, diabetes still affects millions of individuals worldwide, with a high death and disability rate. There has been a concerted search for alternative natural medicinal agents; luteolin (LUT), a polyphenolic molecule, might be a good choice, both because of its efficacy and because of it having fewer side effects, compared to conventional medicines. This study aims to explore the antidiabetic potential of LUT in diabetic rats, induced by streptozotocin (STZ; 50 mg/kg b.w.), intraperitoneally. The level of blood glucose, oral glucose tolerance test (OGTT), body weight, glycated hemoglobin A1c (HbA1c), lipidemic status, antioxidant enzymes, and cytokines were assessed. Also, its action mechanism was explored through molecular docking and molecular dynamics simulations. Oral supplementation of LUT for 21 days resulted in a significant decrease in the blood glucose, oxidative stress, and proinflammatory cytokine levels, and modulated the hyperlipidemia profile. LUT also ameliorated the tested biomarkers of liver and kidney function. In addition, LUT markedly reversed the damage to the pancreas, liver, and kidney cells. Moreover, molecular docking and molecular dynamics simulations revealed excellent antidiabetic behavior of LUT. In conclusion, the current investigation revealed that LUT possesses antidiabetic activity, through the reversing of hyperlipidemia, oxidative stress, and proinflammatory status in diabetic groups. Therefore, LUT might be a good remedy for the management or treatment of diabetes.

## 1. Introduction

Diabetes mellitus (DM) has emerged as a major health issue in recent years, and the complications associated with it are receiving high attention. DM is associated with a deterioration in life expectancy and a worsening in the quality of life. According to the reports of the International Diabetes Federation, the number of adults living with diabetes is 537 million, and estimated to reach 783 million globally, and 124.87 million in India, by 2045 [1,2]. Based on the etiology, diabetes mellitus can be divided into two main types: type 1 and type 2. Type 1 is driven by an autoimmune disease that destroys β-cells, while type 2 is driven by insulin resistance. An upheaval in insulin secretion and action leads to abnormalities in the metabolism of proteins, fats, and carbohydrates, resulting in hyperglycemia, hyperlipidemia, inflammation, and oxidative stress that are implicated in the onset and progression of complications in diabetes [3,4]. To delay, and/or restrict, the progression of diabetes, synthetic pharmacological compounds are employed; however, these compounds are excessively expensive and have various negative effects [5].

Despite the widespread use of oral antidiabetic agents, diabetic complications induced by metabolic alteration, oxidative stress, and inflammation still occur, requiring adjuvant therapies. Immunosugars are the chemical compounds in which nitrogen molecules from the ring have been substituted with oxygen molecules; different immunosugars have been investigated to manage hyperglycemia, for effective management of this ever increasing disorder. These immunosugars were found to reduce intestinal glucose absorption by inhibiting α-1,4-glucosidases enzyme [6]. Although the majority of synthetic oral antidiabetic medications are used to manage the development of diabetes, they only partially reverse the course of its difficulties, and certainly make it worse, because they also exhibit substantial adverse effects [7].

In order to provide novel, effective medications, with fewer side effects, the pursuit of antidiabetic medicines has been directed to medicinal plants [8]. Plant-derived phytoconstituents have received much attention, and are frequently utilized in the primary healthcare setup of many developed and developing countries to treat various diseases and disorders, including diabetes, due to their minimal or limited side effects compared to synthetic drugs [8,9]. LUT’s (polyphenol) structure is given in Figure 1, it is a naturally occurring compound found in many plant-based pharmaceuticals and neutraceuticals [10].

LUT has been shown to have a variety of bioactivities including anti-inflammatory, and antioxidant activity, which may be useful in the treatment of various chronic diseases associated with oxo-inflammations such as cardiovascular disease, liver disease, etc. [11].

Moreover, it also modulates various signaling pathways, gene expression, and cellular metabolism. This makes it a potentially multifunctional molecule that might be used to target many bodily pathways and processes. Besides that, LUT is largely regarded as a safe molecule because it is derived from natural sources [12].

The use of molecular docking and molecular dynamics simulations is a newly developed interdisciplinary approach for high throughput screenings of biomolecules [13]. The molecular docking approach outlines the insertion of a test molecule in the binding site of the receptor of interest. The beneficial binding strength between the ligand and the receptor complex must be predictable using a docking approach. Day by day, initial drug testing increasingly depends on computational chemistry and chemoinformatics [14]. Moreover, in the present study, we performed molecular docking and molecular dynamics (MD) simulations of LUT with key enzymes linked to diabetes. Development of a novel therapeutic candidate is a costly and long process, a combination of molecular docking and MD simulations, in addition to experimental studies, serves as a promising alternative.

Overall, the potential health benefits, natural origin, mechanisms of action, and safety profile of LUT make it a promising candidate for further research; therefore, by using this unique approach, we want to explore the potential of LUT as a safe and effective drug for the management of diabetes.

## 2. Results

### 2.1. Effect of LUT on α-glucosidase

The IC50 value of LUT showed potential inhibitory effects on α-glucosidase in a dose-dependent manner. For glucosidase, the IC50 value was observed at 41.22 ± 1.18. As a result, inhibiting α-glucosidase is a very effective approach to delaying glucose absorption and lowering postprandial blood glucose levels, perhaps slowing diabetes progression.

### 2.2. Oral Glucose Tolerance Test (OGTT)

Hyperglycemia is the most important symptom of diabetes. The OGTT is an important indicator of diabetes alleviation. In the OGTT, the plasma glucose level was taken at 60 and 120 min; a significant reduction in plasma glucose level (glucose load) was observed in the treatment group Figure 2a. The lowering of glucose levels revealed a higher glucose tolerance: either β-cells produced enough insulin or there was higher glucose uptake by tissues in the treated groups. 

Effect of LUT treatment on blood glucose levels in diabetic rats.

Diabetic rats exhibited a significant increase in the levels of blood glucose as compared to the NC group. After administration of LUT to diabetic rats, their blood glucose levels were significantly reduced. The glibenclamide (GLB)-treated group showed similar results to those treated with LUT. The overall results are presented in Figure 2b.

The observed data reveal that LUT treatment might reverse glucose levels in diabetic rats.

### 2.3. Effect of LUT Treatment on HbA1c Level in Diabetic Rats

The glycated Hb level was evaluated to check the efficacy of the test drug for its long-term effect on glucose regulation. As compared to the control group, the HbA1c level was significantly increased in the diabetic group (Figure 1c). After treatment with the test drug and standard, the level of HbA1c returned to a near normal level. The test and standard drug showed an almost similar ameliorative effect.

### 2.4. Body Weight

As shown in Table 1, the initial body weights of all the experimental groups were statistically similar. In the NC, LUT, and GLB groups, the gross body weight of the rats increased significantly, whereas in STZ-induced diabetic rats, a significant loss in body weight was observed, in comparison to the NC and treated groups.

### 2.5. Effect of LUT on Serum Lipid Profiles in Diabetic Rats

Body weight, and different biochemical parameters investigated in this study, revealed the overall level of triglycerides (TG), total cholesterol (TC), high-density lipoproteins (HDL), and low-density lipoproteins (LDL) in the NC, DC, and treatment groups. In the DC group, the levels of TC, TG, and LDL increased significantly, while lower levels of HDL were observed. After oral administration of LUT and GLB, the levels of TC, TG, LDL, and HDL were significantly reduced, returning to near normal levels. The LUT treatment showed a similar ameliorative effect to GLB. Overall, these results indicate that LUT supplementation could improve the hyperlipidemic condition in diabetic rats (Table 1).

### 2.6. Effect of LUT on Liver Function Tests in Diabetic Rats

As compared to the NC group, significant increases in the alanine transaminase (ALT), aspartate transaminase (AST), and alkaline phosphatase (ALP) levels were observed in the DC group. After treatment with LUT, a remarkable reduction in the levels of ALT, AST, and ALP was observed. After 21 days of treatment with LUT and GLB, the levels of these markers was nearer to the values of the NC group (Table 2).

LUT and GLB showed similar results, except for ALT. The results suggest that LUT could modulate liver function.

### 2.7. Effect of LUT on Kidney Function Tests in Diabetic Rats

The results of kidney function tests of all the groups are shown in Table 2. The DC group had severely impaired kidney function. After treatment with LUT, the increased levels of urea, creatinine, and uric acid were reversed, to normal levels. LUT showed better ameliorative results in the case of urea and uric acid, while GLB showed similar results for all the parameters. In the case of urea and uric acid, both LUT and GLB showed similar results, whereas in the case of creatinine, GLB showed better results as compared to LUT. However, both LUT and GLB treatments could significantly improve kidney function.

### 2.8. Effect of LUT on Antioxidant Status

The results of SOD, CAT, and GSH expression in the groups are presented in Table 2. In the DC group, the expression of SOD, CAT, and GSH was significantly decreased compared to the NC group. After treatment with LUT and GLB, the expression of SOD, CAT, and GSH was significantly increased, returning to a near normal level. In the case of SOD and GSH, similar results were recorded for both LUT and GLB, while in the case of CAT, slightly better results were found for GLB. These results indicate that LUT might alleviate DM through increasing the levels of antioxidant enzymes.

### 2.9. Effect of LUT on TNF-α and IL-6 Expression Levels in Serum

The expression of tumor necrosis factor-α (TNF-α) in all the groups was analysed using ELISA. Table 2 reveals that TNF-α expression recorded in the DC group was about 3.5-fold higher compared to the NC group, revealing an obvious inflammatory condition. After treatment with LUT and GLB, the expression of TNF-α significantly decreased, returning to a near normal level. Similar results were observed in the case of interleukin 6 (IL-6). Treatment with LUT suppressed the exhibited levels of these proinflammatory factors, suggesting that LUT is effective in reducing the inflammation associated with diabetes.

### 2.10. Histopathology of the Pancreas, Liver, and Kidneys

Histological observation of pancreatic cells showed circumscribed masses surrounded by deeply stained pancreatic exocrine cells, and β-cells are identified in the NC group. In the DC group, β-cells degenerated and appeared disorganized, as shown in Figure 3.

In the treatment group, the population of β-cells and pancreatic tissues were significantly restored, near to normal, as evidenced by the histological observation. The histological observation of the liver section of the STZ-induced DC group showed congestion of hepatocellular necrosis, and central veins with vacuolization, compared to the NC group (Figure 4).

However, after treatment with LUT and GLB, a normal hepatic architecture was attained by reversing hepatic necrosis, cell vacuolization, and central veins. Histological observation of the kidney cells revealed that the NC group displayed normal glomerulus and Bowman’s capsules, while in the DC group, abnormal glomerular architecture and dilated Bowman’s capsules were observed. LUT and GLB treatment returned the glomerulus to normal and regulated the Bowman’s capsules (Figure 5).

### 2.11. Molecular Docking

To explore the possible binding interactions between LUT and the antidiabetic receptor glucosidase, we carried out molecular docking studies of the LUT into the native ligand binding regions of the α-glucosidase receptor (PDB: 3W37) Table 3.

Docking scores, binding energies, and the types of interaction of LUT with α-glucosidase are presented. As shown in Figure 5, acarbose interacted with residues His626, Asp357, Asp630, Asp232, Glu630, Asp568, Arg552, and Phe601 via hydrogen bonds, and Trp432, Trp329, and Asp469 via hydrophobic interactions. LUT interacts with the α-glucosidase receptor using interactive residues ASP 232, Asp552, and Asp357 through hydrogen bonding interactions (Figure 5a). A superimpose of the docked pose of LUT with acarbose (orange colour stick model) at the binding site of α-glucosidase is given in Figure 6b.

The LUT compound showed a docking score of −7.701 kcal/mol and glide energy of −54.698 kcal/mol (Table 3). These three amino acid residues showed the same hydrogen bonding interaction as the standard drug acarbose. The oxygen group and 7-OH group of the 4H-chromene moiety formed a hydrogen bonding interaction with the NH group of Asp552 (3.34 Å), and the CO group of Asp232 (1.76 Å), respectively, and the 2-OH group of the benzene-1,2-diol moiety also interacts through hydrogen bonding with the O atom of Asp357 (1.89 Å). The other amino acid residues, such as His626, Asp469, Phe601, Asp568, Glu603, Asp630, Trp432, and Trp329, showed nonbonded hydrophobic interactions. The binding interaction and surface cavity of LUT are given in Figure 7.

### 2.12. Molecular Dynamics Simulation

The root-mean-square deviation (RMSD) of the MD simulation trajectories system was employed to examine the dynamic profile of the α-glucosidase receptors. The RMSD suggest that docked ligand complex systems attained equilibrium within a time interval of 10,000 picoseconds (ps), and remained stable throughout the MD simulation. The ligands’ RMSD plot indicates that acarbose exhibits lower RMSD values than LUT (Figure 8a,b).

The MD simulation shows different types of bonding between ligand receptors, 2D ligand receptors and the percentage of amino acid interaction are shown in Figure 9.

Two-dimensional ligand–protein interactions in the molecular dynamics simulation revealed that acarbose and LUT form hydrogen bonds, hydrophobic contacts, and water bridges. In both acarbose and LUT, the hydrogen bonds predominantly bind with different amino acids, to attain equilibrium state. For each ligand, the RMSD of the ligand with respect to the protein backbone was further examined; the RMSDs of the 3W37 complex with LUT and acarbose were 4.8 ± 1.1 and 1.1 ± 0.4, respectively.

## 3. Discussion

LUT is a polyphenol used to treat various diseases and disorders by influencing oxidative stress, inflammation, and dyslipidemia, and slowing carbohydrate digestion and absorption by interacting with α-glucosidase [15,16]. In this study, we showed for the first time that LUT displayed adequate antidiabetic effects in diabetic rats, and also explored the mechanism of action using molecular docking and a molecular dynamics simulation. Diabetes induced by STZ, results in altered glucose levels, a proinflammatory state, a lipidemic state, and a redox imbalance, which are characteristics of diabetes mellitus. LUT supplementation for three weeks in diabetic rats significantly attenuated the hyperglycemia, HbA1c, hyperlipidemia, inflammation, and antioxidant enzyme activity [17,18].

Previously published reports demonstrated that loss in body weight is a typical characteristic of DM, which was evidently shown in our experiment [19]. Body weight loss in the DC group occurs due to the atrophy of structural proteins and muscle wasting. Oral administration of LUT significantly attenuated the weight loss, returning rats to near normal weight, suggesting that improvement of protein synthesis, as a result of glycemic control, ultimately improves the quality of life in diabetic conditions [20].

Blood glucose values and OGTT are important indicators, and are considered key symptoms of DM [21]. In this research, LUT administration to diabetic rats markedly reduced blood glucose levels and significantly improved the OGTT. These results indicate that LUT has a possible hypoglycemic effect. This could be a result of LUT’s ability to regenerate pancreatic β-cells and secrete insulin, or a result of its ability to stimulate the release of bound insulin from β-cells by inhibiting ATP-sensitive K+ channels, similar to GLB [22]. The HbA1c assay is the most reliable marker to assess the efficacy of the treatment for long-term blood glucose control. The non-enzymatic attachment of glucose to the N-terminal valine residue of hemoglobin, results in the formation of glycated HbA1c. In addition, the degree of glycation relies on the blood glucose in the biosystem [23]. The hypoglycemic effect of the standard drug GLB was evident due to the stimulation of insulin release from the pancreas, and inhibition of glucagon secretion. The LUT treatment for diabetic rats possessed insulin-secretory effects like GLB [24]. These findings showed that LUT can reduce insulin resistance in diabetic animals [25,26]. This was further substantiated by histology findings, which showed that diabetic groups displayed shrinkage, necrosis, and damaged β-cell populations; however, after administering LUT, these effects returned to normal. We found an interesting agreement between our findings and recently published reports [25,26].

Lipidemic dysfunction is typically linked with DM, and plays a crucial role in the development of cardiometabolic disorders, hence, as a result of lipidemic malfunction, diabetic patients are more prone to cardiovascular diseases. Similarly, in the current investigation, a close relation between hyperglycemia and dyslipidemia was recorded [27]. This is recognized as a complication of DM, owing to the increased breakdown of lipids and free fatty acids from peripheral deposits in insulin deficiency [28]. STZ markedly increased the levels of TC, TG, and LDL, but decreased HDL levels in diabetic rats. LUT effectively reduced the levels of TC, TG, and LDL, while elevating the HDL level in diabetic rats. These obtained results may be the result of inadequate insulin production and secretion during a hyperglycemic condition, which could trigger free fatty acid mobilization from adipose tissue through the action of hormone-sensitive lipase [26]. The lower lipid action shown in the LUT treatment might be due to relative gene expression, and may alleviate dyslipidemia, inhibit lipid synthesis, and further improve diabetes [29].

Both the liver and kidneys are important organs in the body, that play a crucial role in regulating many physiological processes. Damage to these organs, either by oxidative stress, inflammation, or hyperlipidemia, can have grave consequences for human health [30]. The liver regulates blood glucose levels by storing glucose as glycogen, converting it to glucose as needed, and producing glucose from non-carbohydrate sources such as amino acids. Thus, in cases of liver damage, the levels of ALT, AST, and ALP increase sharply in diabetic groups, indicating the severity of the liver injury, which suggests that impaired liver function could be mediated by hyperglycemia [3,30]. LUT treatment significantly modulates the levels of these markers, similar to GLB. Therefore, the restoration of these markers to normal levels denotes a reduction in diabetic complications. In addition, the markers of renal damage such as urea, uric acid, and creatinine were measured in order to assess the functional efficiency of the kidney. In the present investigation, the levels of these biomarkers were found to be increased in the diabetic groups, while being restored to normal after LUT administration, which is in agreement with previous reports [31]. The restoration of the kidney biomarkers suggests that LUT has the ability to reduce renal impairment in diabetic complications, through the inhibition of protein degradation associated with DM. Histological examinations of the liver and kidneys of the diabetic group showed damaged hepatocytes and renal cells, respectively. Treatment with LUT revealed a remarkable improvement in the aforesaid histoarchitectural alterations, which indicates the healing and rejuvenating potential of LUT.

Humans have an effective defense mechanism against free-radical-induced damage, using antioxidant enzymes like SOD, CAT, and GSH. Oxidative stress (caused by abnormal levels of free radicals or a decline in antioxidant enzymes) can lead to diabetes as a chronic ailment [24,31,32]. In the present investigation, reduced activity of SOD, CAT, and GSH was shown in diabetic groups, whereas, after treatment with LUT, these levels were significantly restored to normal. In addition, these enzymes play a mutually supportive role in defense against reactive oxygen species (ROS).

Notably, the previously reported literature has suggested that abnormal cytokine (TNF-α and IL-6) levels in the blood are associated with pancreatic β-cell death and dysfunction [31,33]. In the recent past, research has shifted the viewpoint on diabetes, from it being a metabolic disease to an inflammatory condition. To maintain the normal physiological function of pancreatic β-cells, the proinflammatory cytokines need to be balanced [33]. In this investigation, a significant reduction in the levels of TNF-α and IL-6 was observed in the case of LUT treatment, while opposite results were recorded in the diabetic groups. LUT and the standard GLB showed very similar ameliorative results. This mechanism may be useful in ameliorating β-cell dysfunction, which is the hallmark of DM.

The in silico molecular studies of LUT’s interactions with the active sites of the enzyme (glucosidase), showed excellent inhibition of enzyme activity. The interactions between proteins and ligands showed strong hydrogen bonding. These confirm that LUT can be used as an antidiabetic candidate [34]. Polyphenols modulate diabetes by hindering the digestion and glucose absorption of carbohydrates, by interacting with the enzyme α-glucosidase. According to Villa-Rodriguez et al., bioactive compounds are effective in attenuating carbohydrate digestion and sugar absorption; it is reported that plants containing polyphenols are very beneficial in attenuating digestion and absorption of carbohydrates [35]. The substantial binding between ligands and proteins in the docking studies was further supported by the MD simulation experiment. Acarbose and LUT were found to have the lowest RMSD values, and the other metrics confirmed the stable binding between the native structure and ligand complex.

Luteolin (LUT) showed an excellent antidiabetic effect in the present investigation and it is very similar to the standard, GLB. Therefore, LUT could be deemed as a good candidate, in the future, for the management or treatment of diabetes. Although this work reports on the antidiabetic activity of LUT, systematic investigations should be carried out both on its effects and mechanism, as well as its toxicity, to establish its actual role in diabetes.

## 4. Materials and Methods

The commercial kits for the measurements of TC, TG, low-density lipoprotein LDL, high-density lipoprotein HDL, TNF-α, IL-6, and HbA1c were purchased from Sigma-Aldrich. Luteolin 97% was procured from Almanac Life Science India Pvt. Ltd., New Delhi, India. All the reagents were of analytical grade.

### 4.1. In Vitro α-glucosidase Activity

In vitro α-glucosidase activity was evaluated according to the defined protocol [36,37]. Different concentrations of selected compounds (33–500 µg/mL) were used to investigate the α-glucosidase inhibition potential. Acarbose was used as a reference compound positive control. All of the tests were carried out in triplicate. The equation below was used to express the results.
$$Percentage=\frac{A\;control-A\;sample}{A\;control}$$

### 4.2. Animals

Wistar albino rats, weighing about 150 g, were acquired from the Central Animal House at Jamia Hamdard in New Delhi, India. The animal procedures were performed as per CCSEA guidelines. According to the design protocol, animals were divided into different groups (n = 6). A 12 h cycle of light and dark was maintained at 25 ± 2 °C. Experimental animals were familiarized for a week prior to starting the experimentation, and normal chow was given.

#### 4.2.1. Induction of Type 2 Diabetes and Grouping of Animals

A total of 24 animals were taken in this study, all the animals were divided into four groups. Group 1, received the vehicle, group 2 received a toxicant (STZ 50 mg/kg), group 3 received LUT (100 mg/kg per oral), and group 4 received glibenclamide (5 mg/kg per oral). After an STZ injection, fasting blood glucose levels were measured. Only animals having blood glucose levels higher than 250 mg/dL were included in the study. Blood was collected from the retro-orbital plexus. Animals were sacrificed and disposed of by biotic waste management, New Delhi, as per CCSEA guidelines [20].

#### 4.2.2. Oral Glucose Tolerance Test (OGTT)

A test for OGTT was performed at an early stage in the experiment. The level of blood glucose 1 was measured at 0, 30, 60, and 120 min after oral administration of glucose (2 g/kg body weight, 2% *w*/*v* glucose solution). Blood sugar levels were measured using a glucometer [3].

#### 4.2.3. Blood Glucose Level

Blood glucose in all experimental groups was documented on day 0, 7, 14, and 21 after 12 h fasting, and was checked using a strip glucometer.

#### 4.2.4. Glycated Hemoglobin Level

Glycated hemoglobin A1c (HbA1c) was measured in all treatment groups according to the manufacturer’s protocol using a span diagnostic kit.

### 4.3. Body Weight

Body weight was measured on day 0, 7, 14, and 21.

### 4.4. Biochemical Analysis

The rats were anaesthetized and blood was collected from the eyeball of the rat. Further, serum was recovered through centrifugation at 3000× *g* for 10 min and stored at −80 °C until use. Serum glucose level, liver function parameters (ALT, AST, ALP), kidney function parameters (urea, creatinine, uric acid), and lipid profiles (TG, TC, LDL-C, and HDL-C) were assayed [3,32].

### 4.5. Detection of TNF-α and IL-6 Levels in Serum

Quantification of the levels of TNF-α and IL-6 expression in the serum of all the groups of rats under study was performed by ELISA kit. The obtained results were expressed as pg/mL [3].

### 4.6. In Vivo Antioxidant Activity

After the experiment was completed, a piece of the liver was homogenized and processed in PBS to determine SOD, CAT, and GSH activity/levels [32].

### 4.7. Histopathological Studies

For histological analysis, 10% formalin was utilized as a fixative and paraffin was used to embed cells from the pancreas, liver, and kidneys from experimental animals. Hematoxylin and eosin (H&E) staining was performed on tissue slices cut at a thickness of 5 mm staining agent [3,32].

### 4.8. Molecular Docking

Our docking study was conducted using Maestro Schrodinger (version 11.8) to determine the modes of interaction of LUT in the active sites of α-glucosidase [38]. We selected chain D for the 3W37 protein ID, due to the acarbose moiety, and deleted the other chain along with residual water molecules that were on the far side 5 Å, thus producing an energy-minimized structure. Grids were generated using these energy-minimizing protein structures along with an acarbose ligand, to represent the binding sites within the receptor target. The docking was completed by utilizing the Glide tool, with greater accuracy and writing XP descriptor data. Using this method, we produced advantageous ligand poses to establish their spatial match into receptor active sites, and then we evaluated and minimized the best-fitting poses. The docking scores, glide energy, hydrogen bonds, and the pi–pi interactions created with the circumferential amino acids of these compounds were utilized to estimate the binding affinity and appropriate alliance of these compounds in the active site of the receptor or enzyme.

### 4.9. Molecular Dynamics Simulation

The Desmond tool, from Schrodinger, was used to run MD simulations to obtain an insight into the binding stabilization of the receptor α-glucosidase’s (PDB: 3W37) native structure and in docked complexes with LUT and acarbose. Before the MD simulations, the hydrogen bond network in both the native and docked complex structures were refined at pH, employing the Protein Preparation Wizard from Schrodinger, and the final restrained minimization was carried out using the OPLS3e force field [39]. An orthorhombic box, solvated with a TIP3P water model, 0.15 M NaCl counter ions, and a system builder module, were used to introduce more minimized structures into the workspace. Before the MD simulation, all prepared systems were relaxed by a series of energy minimization and brief MD simulations. In addition, MD simulations were then run for a total of 10,000 picoseconds (ps) for each approach, saving the coordinates every 10 ps at 300 K and 1.0325 bar pressure. The Simulation Event Analysis module in Desmond was used to further evaluate the simulation findings using the RMSD.

### 4.10. Statistical Analysis

Data were examined using Prism in Graph Pad. All the results are displayed as the mean ± SD (standard deviation). Statistics were considered significant for *p* values < 0.05.

## 5. Conclusions

LUT, a naturally occurring polyphenolic, has been shown to possess potential health benefits for the management of diabetes. The current investigation revealed that LUT can reduce blood glucose levels, lipid malfunction, and body weight in diabetic rats, and also delay carbohydrate digestion and absorption. Moreover, the reversal of oxidative stress and inflammatory status confirms its antioxidant and anti-inflammatory effects. Further, the in vivo results were supported by molecular modelling and simulations, molecular docking with the 3W37 protein was shown to have comparable results to the innate ligand acarbose. In conclusion, LUT represents a potentially effective alternative for the development of novel therapeutics to manage diabetes, and may enhance the quality of life for those who already have the disease. However, this investigation needs extensive investigation on a molecular level to confirm its antidiabetic potential.

## Figures and Tables

**Figure 1 jfb-14-00126-f001:**
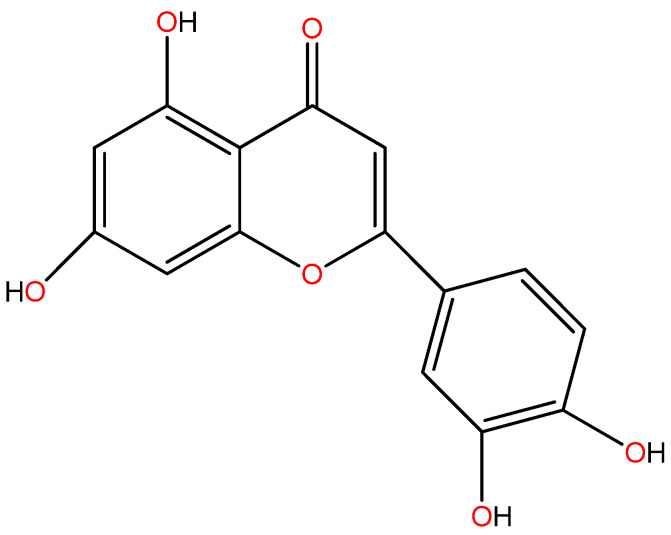
Molecular structure of LUT.

**Figure 2 jfb-14-00126-f002:**
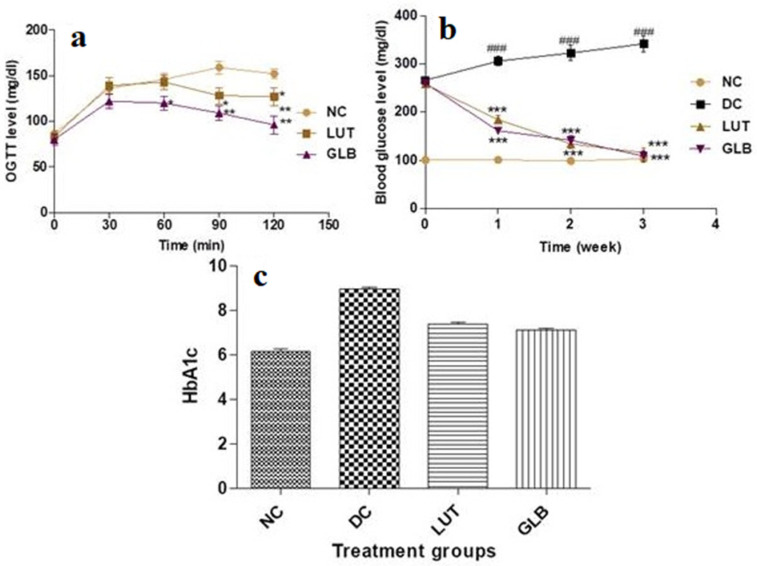
Oral glucose tolerance test (OGTT) in normoglycemic rats (**a**). Blood glucose levels in experimental groups of rats (**b**). HbA1c levels in experimental groups of rats (**c**). Normal control (NC), diabetic control (DC), treatment (LUT), and glibenclamide (GLB). Data are expressed as mean ± SD (N = 6). Values with superscripts (*, **, *** ) are significantly (*p* < 0.05, 0.01, 0.001, respectively) different from the DC group (#, ##, ###, respectively).

**Figure 3 jfb-14-00126-f003:**
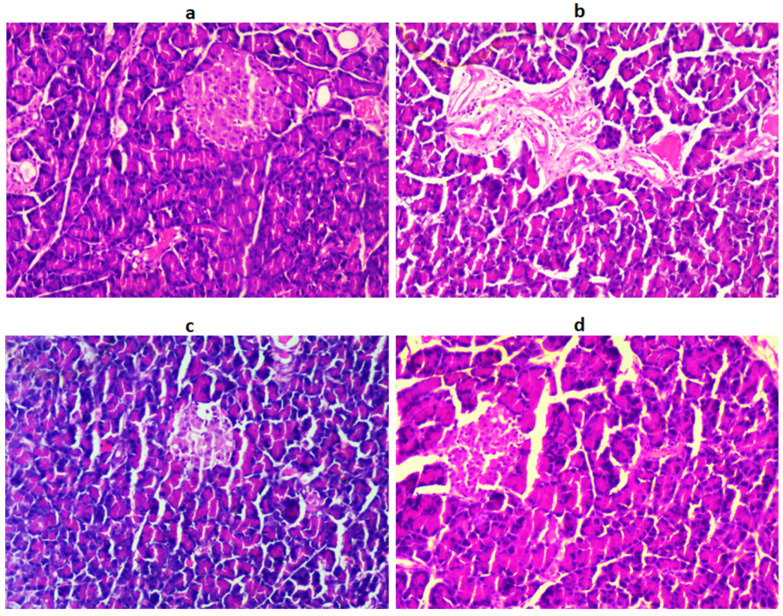
The histological architecture of the pancreatic tissues in the DC group (**b**) shows destructed pancreatic exocrine cells and islets of β-cells with congested blood capillaries. After treatment with LUT (**c**) and standard (**d**), the histology of the pancreas was reversed to normal (**a**).

**Figure 4 jfb-14-00126-f004:**
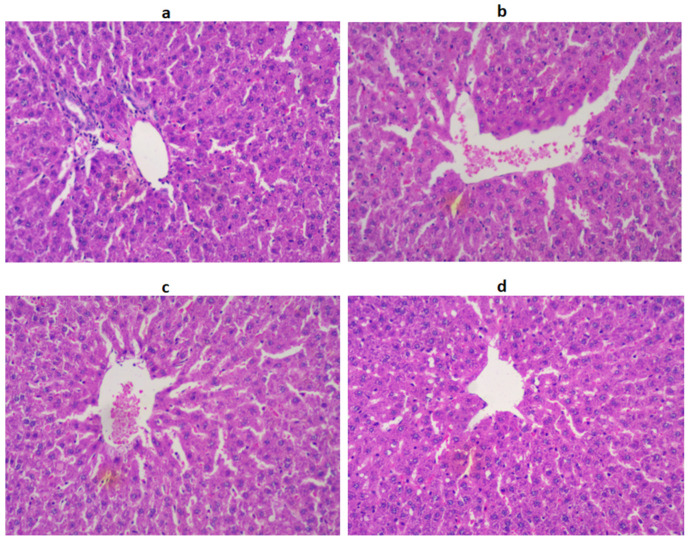
The histological architecture of the liver tissues in the DC group (**b**) shows degenerated cells with fading nuclei and irregular hepatic cords, with the disarranged CV. After treatment with LUT (**c**) and standar (**d**), the histology of the liver was returned to normal (**a**).

**Figure 5 jfb-14-00126-f005:**
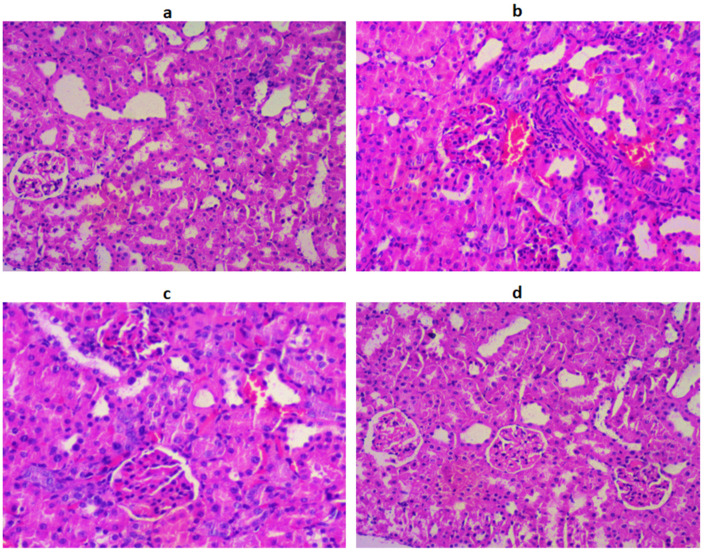
The The histological architecture of the kidney tissues in the DC group (**b**) is damaged, and cell infiltration and interstitial hemorrhaging are found, along with destructured brush border cells and missing Bowman’s capsules. After treatment with LUT (**c**) and standard (**d**), the normal histology of kidney tissues was observed as in the NC group (**a**).

**Figure 6 jfb-14-00126-f006:**
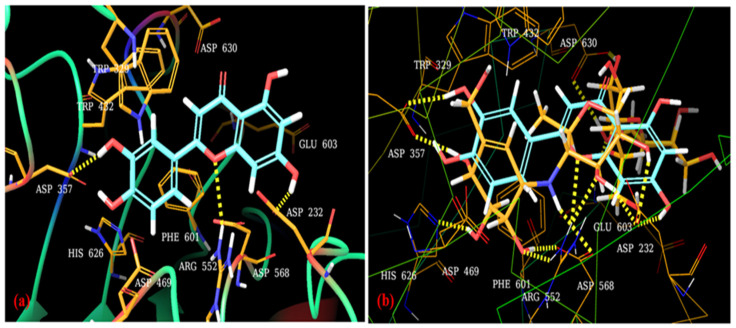
(**a**) Docked pose of LUT (turquoise color, stick model) at the binding site of α-glucosidase (PDB: 3W37) (**b**) Superimpose of the docked pose of LUT with innate ligand acarbose (orange color, stick model) at the binding site of α-glucosidase (PDB: 3W37). Hydrogen bonds are shown as yellow dashed lines.

**Figure 7 jfb-14-00126-f007:**
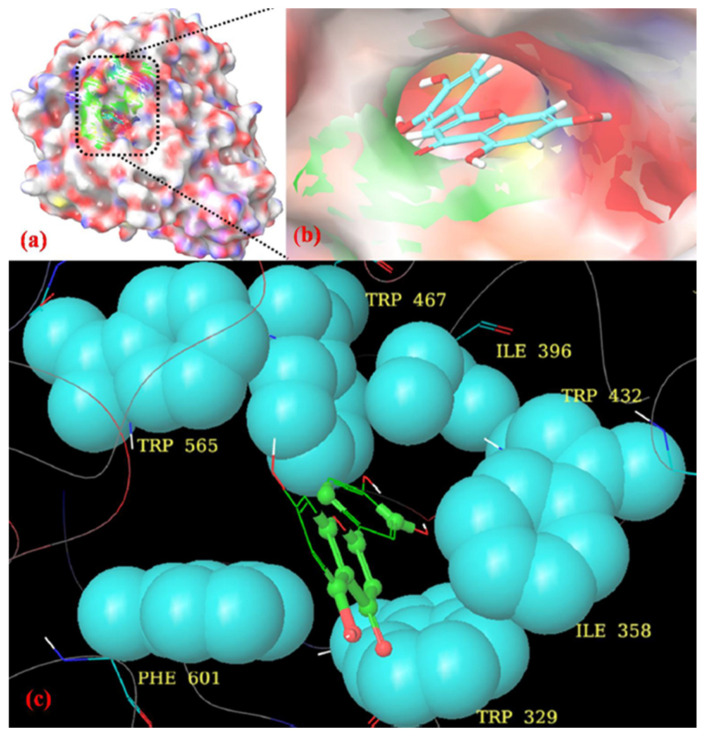
(**a**) Molecular surface view of LUT, represented in turquoise color stick model, at the binding site of α-glucosidase (PDB: 3W37). (**b**) Zoomed in view of LUT. (**c**) Hydrophobic enclosure of LUT. The hydrogen bond is shown as a yellow dashed line.

**Figure 8 jfb-14-00126-f008:**
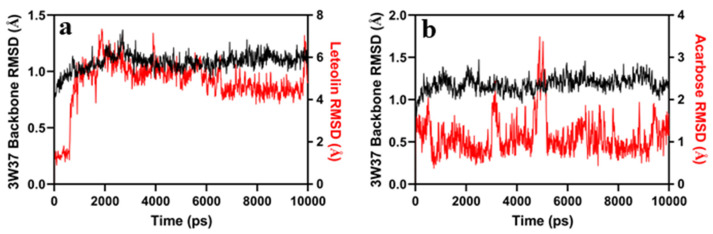
Molecular dynamics simulation of α-glucosidase receptor (PDB: 3W37), native, and docked complex with LUT, and acarbose. (**a**) RMSD of docked complex 3W37 backbone with ligand LUT, plotted against time (ps). (**b**) RMSD of docked complex 3W37 backbone with innate ligand acarbose, plotted against time (ps).

**Figure 9 jfb-14-00126-f009:**
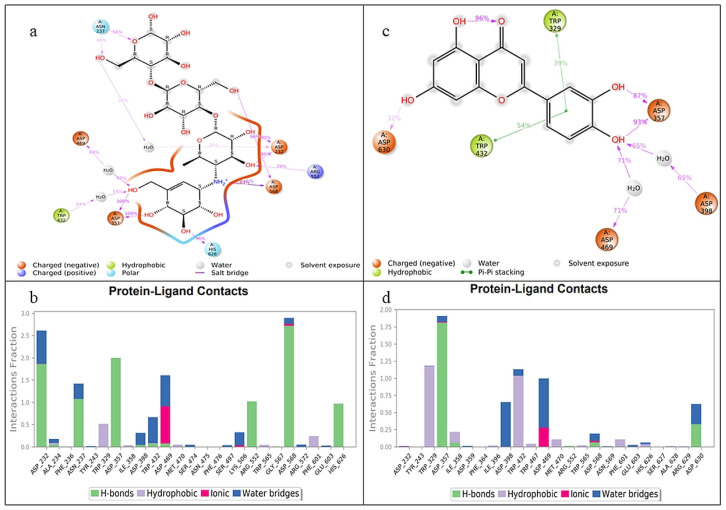
(**a**) Two-dimensional image of ligand (acarbose) percentage interaction with amino acids of α-glucosidase (3W37) receptor. (**b**) Bar chart of amino acids’ interaction with ligand (acarbose). (**c**) Two-dimensional image of ligand (LUT) percentage interaction with amino acids of α-glucosidase (3W37) receptor. (**d**) Bar chart of amino acids’ interaction with ligand (LUT).

**Table 1 jfb-14-00126-t001:** Body weight and different biochemical parameters (TG, TC, HDL, and LDL) in NC, DC, GLB, and treatment groups.

Tests Parameters		Treatment Groups				
Body weight		Treatment Day	NC	DC	LUT	GLB
		Day 0	161.87 ± 4.86	164.37 ± 5.24 ^ns^	160.68 ± 3.73 ^ns^	161.13 ± 5.94 ^ns^
		Day 7	171.85 ± 3.27	154.78 ± 4.97 ^ɸ^	164.18 ± 4.21 ^€^	166.46 ± 3.55 ^£^
		Day 14	184.40 ± 3.44	141.52 ± 5.85 ^ɸ^	180.19 ± 5.71 ^£^	176.87 ± 5.12 ^£^
		Day 21	200.80 ± 3.96	132.00 ± 5.96 ^ɸ^	194.07 ± 4.75 ^£^	202. 09 ± 5.02 ^£^
Lipid profile	TC (mg/dL)	Day 0	78.78 ± 2.44	183.38 ± 4.42 ^ɸ^	95.79 ± 3.86 ^£^	89.38 ± 3.00 ^£^
	TG (mg/dL)	Day 7	111.75 ± 2.36	164.10 ± 2.43 ^ɸ^	117.13 ± 4.31 ^€^	107.05 ± 4.40 ^£^
	HDL (mg/dL)	Day 14	30.14 ± 0.98	22.68 ± 1.11 ^ɸ^	27.34 ± 0.57 ^€^	27.18 ± 1.21 ^£^
	LDL (mg/dL)	Day 21	30.40 ± 1.51	103.42 ± 2.90 ^ɸ^	46.95 ± 2.55 ^£^	36.51 ± 2.51 ^£^

Values indicate mean ± SD (n = 6). $/ϕ *p* < 0.05, ^€^/ʠ *p* < 0.01, and ^£^/^ɸ^ *p* < 0.001, compared with treatment and normal control, respectively. ns = not significant.

**Table 2 jfb-14-00126-t002:** Different biochemical parameters (LFT, KFT, SOD, CAT, GSH, TNF, and IL-6) of NC, DC, GLB, and treatment groups’ animals.

Biochemical Parameters	Treatment Groups				
Liver function tests		NC	DC	LUT	GLB
	AST (U/L)	51.72 ± 1.01	98.24 ± 2.78 ^ɸ^	68.08 ± 2.83 ^£^	59.71 ± 2.25 ^£^
	ALT (U/L)	68.14 ± 2.06	111.06 ± 4.95 ^ɸ^	95.23 ± 4.50 ^€^	70.75 ± 3.87 ^£^
	ALP (U/L)	117.63 ± 3.44	213.30 ± 5.63 ^ɸ^	147.41 ± 3.51 ^£^	142.59 ± 3.50 ^£^
Kidney function tests	Serum urea (mg/dL)	38.70 ± 1.36	68.54 ± 2.43 ^ɸ^	46.40 ± 1.48 ^£^	41.20 ± 1.16 ^£^
	Serum creatinine (mg/dL)	0.91 ± 0.06	1.34 ± 0.06 ^ɸ^	1.01 ± 0.07 ^€^	0.92 ± 0.07 ^£^
	Uric acid (mg/dL)	8.41 ± 0.23	19.05 ± 0.96 ^ɸ^	10.66 ± 0.56 ^£^	8.56 ± 0.89 ^£^
Antioxidant status	SOD (U/mg protein)	9.10 ± 0.09	5.82 ± 0.11 ^ɸ^	7.22 ± 0.09 ^£^	8.83 ± 0.06 ^£^
	CAT (U/mg protein)	66.37 ± 2.33	47.07 ± 2.61 ^ɸ^	57.24 ± 2.65 ^€^	60.80 ± 1.72 ^£^
	GSH (U/mg protein)	12.73 ± 0.87	6.27 ± 0.80 ^ɸ^	9.27 ± 0.93 ^€^	9.63 ± 0.92 ^€^
Antioxidant status	TNF-α	52.28 ± 3.47	179.08 ± 7.54 ^ɸ^	82.95 ± 3.17 ^£^	87.77 ± 3.18 ^£^
	IL-6	40.42 ± 3.29	108.75 ± 5.14 ^ɸ^	68.19 ± 2.91 ^£^	71.31 ± 3.01 ^£^

Values indicate mean ± SD (n = 6). $/ϕ *p* < 0.05, ^€^/ʠ *p* < 0.01, and ^£^/^ɸ^ *p* < 0.001, compared with treatment and normal control, respectively. ns = not significant.

**Table 3 jfb-14-00126-t003:** Docking scores, binding energy, and type of interaction of LUT at the active sites of α-glucosidase receptor (PDB: 3W37).

Protein (PDB)	Docking Score	Glide Energy (Kcal/mol)		Hydrogen Bond Interaction		
3W37	−7.701	−54.698	Atom of Ligand	Atom of Amino acids	Amino acids	Dist. (Å)
			ASP 232	H	O	1.76
			ASP 552	O	H	3.34
			ASP 357	H	O	1.89

## Data Availability

As mentioned in table and figures.

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
