# Peer review of "Evaluation of Antidiabetic Effect of Luteolin in STZ Induced Diabetic Rats: Molecular Docking, Molecular Dynamics, In Vitro and In Vivo Studies"

_jfb, 2023, doi:10.3390/jfb14030126_

Round 1
Reviewer 1 Report
Authors report on an antidiabetic action of luteolin as tested on streptozotocin induced diabetic rats. Effect of luteolin was monitored on alpha-glucosides, glucose levels, on glycosylated Hb levels, on serum lipids, on selected organs etc supported by molecular docking study and molecular dynamics simulation. Overall, manuscript is very interesting and gives a detailed overview on effects of luteolin. I strongly support publishing this contribution.
Some minor comments:
- Authors do give some info on luteolin in the introduction section, however, some more info regarding the reasons for choosing it would also be beneficial.
- I don’t find any statement where luteolin was purchased.
- Conclusion is surprisingly short. Authors should extend it.
Author Response
Dear Reviewer
We are appreciating the time and effort that you have dedicated to providing your valuable feedback on my manuscript. We wish to thank you all for your constructive comments on our paper. The comments provided are valuable insights to refine its contents and analysis. Here is a point-by-point response to the comments. We have also incorporated the appropriate changes throughout the manuscript as per the suggestion. All the changes have been incorporated in the revised manuscript with “TRACK CHANGE”.
In this document, we have tried to address all the issues raised by you as best as possible from our side. However, we welcome any further comments for possible improvement of the manuscript.

Reviewer 2 Report
Through this manuscript, the authors wants to address about the controlling of Type 2 diabetes using natural remedies basically the Luteolin which is a natural product. To justify the antidiabetic, effect the authors performed in vivo as well as in sillico studies and the results are found to be rational. This is highly relevant to the readers because natural remedy of T2DM will be highly appropriate than other therapeutics due to probability zero side effects/better availability.
The scope of this topic is high. The paper is presented rationally up to the journal standard.
The abstract and introduction part presented with good scientific languages.
(Note: Structure of the molecule Luteolin should be incorporated in the introduction section)
Although present article emphasis the importance of traditional medicine, it is suggested to mention importance of modern medicine in relation to efforts towards development of antidiabetic medicine. In this regard, it is recommended to emphasis the importance of iminosugars and sugar derivatives as an antidiabetic agents, and it is suggested to cite following relevant articles related to iminosugars in introduction section.
Rajasekaran, P.; Ande, C.; Vankar, Y. D. Synthesis of (5,6 & 6,6)-oxa-oxa annulated sugars as glycosidase inhibitors from 2-formyl galactal using iodocyclization as a key step. ARKIVOC 2022, vi, 5−23.
The experiments methods performed for antidiabetic effect on STZ induced diabetic rats are found to be accurate and used standard protocols. The results for all analysis are found to be in line with each other. For eg. the glycosidic inhibition (IC50= 41.22 ± 1.18) and histological studies clearly suggest the potency of the drug candidate Luteolin.
The molecular docking studies showed the interaction of alpha glycosidase and the ligand Luteolin of similar type (hydrogen bonding) as the drug acarbose and it looks rational and can be compatible with it.
All images are visible and presented in line with the text presented.
Overall, after addressing the points mentioned above, I recommend this article to publish in J. Funct. Biomater.
Author Response
Dear Reviewer,
We are appreciating the time and effort that the editors and reviewers have dedicated to providing their valuable feedback on my manuscript. We wish to thank you all for your constructive comments on our paper. The comments provided are valuable insights to refine its contents and analysis. Here is a point-by-point response to the reviewers’ comments. We have also incorporated the appropriate changes throughout the manuscript as per the suggestion. All the changes have been incorporated in the revised manuscript with “TRACK CHANGE”.
In this document, we have tried to address all the issues raised by the review committee as best as possible from our side. However, we welcome any further comments for possible improvement of the manuscript.

Reviewer 3 Report
As far as I am concerned, this manuscript could be accepted in the present form. In any forthcoming reaserch, try to illuminate the mechanisms underlying the antidiabetic potential.
Author Response
We are appreciating the time and effort that the you have dedicated to providing their valuable feedback on my manuscript. We wish to thank you all for your constructive comments on our paper. The comments provided are valuable insights to refine its contents and analysis. Here is a point-by-point response to your comments. We have also incorporated the appropriate changes throughout the manuscript as per the suggestion. All the changes have been incorporated in the revised manuscript with “TRACK CHANGE”.
In this document, we have tried to address all the issues raised by the review committee as best as possible from our side. However, we welcome any further comments for possible improvement of the manuscript.
